# Recommandation séquentielle pour les articles scientifiques

Julien Aubert-Béduchaud[1]    Florian Boudin[1,2]    Béatrice Daille[1]    Richard Dufour[1]
(1) Nantes Université, École Centrale Nantes, CNRS, LS2N, UMR 6004, F-44000 Nantes, France
(2) Inria, LS2N, Nantes Université, France
`prenom.nom@univ-nantes.fr`

RÉSUMÉ ______________

La croissance rapide du volume de publications scientifiques rend de plus en plus difficile l'identification des travaux clés pour les chercheurs novices. Pour répondre à ce problème, nous proposons une nouvelle tâche de recommandation séquentielle appliquée à la recommandation d'articles scientifiques. Cette formalisation permet de personnaliser les parcours de lecture et de prédire *quoi lire ensuite* à partir de l'expérience d'un chercheur. Nous présentons une étude préliminaire proposant un nouveau jeu de données dédié exploitant les schémas de citation présents dans les introductions des articles comme des chemins de lecture implicites, un protocole d'évaluation adapté aux spécificités de cette tâche, ainsi que des résultats expérimentaux de référence établissant une base de comparaison pour les travaux futurs.

ABSTRACT ______________
Sequential Recommendation for Scholarly Articles

The rapid growth of scientific publications makes it increasingly difficult for novice researchers to identify key works in their field. To address this challenge, we introduce a new sequential recommendation task applied to scientific article recommendation. This formulation enables personalized reading paths and predicts *what to read next* directly from a researcher's experience. We present a preliminary study proposing a dedicated dataset that leverages citation patterns found in article introductions as implicit reading trails, an evaluation protocol tailored to the specificities of this task, and baseline experimental results providing a reference for future work.

MOTS-CLÉS : articles scientifiques, listes de lectures, recommandations séquentielles, séquences de lecture.

KEYWORDS: scientific articles, reading lists, sequential recommendation, reading sequences.

# 1    Introduction

Face à l'explosion récente du volume de publications scientifiques, les chercheurs novices peinent à identifier les articles clés de leur domaine. Les revues de littérature constituent des ressources précieuses pour la montée en compétence, mais elles sont souvent incomplètes, obsolètes ou fragmentaires. Bien que des moteurs académiques comme Google Scholar ou Semantic Scholar facilitent l'identification d'articles pertinents, ils ne cartographient pas les liens entre les travaux et offrent peu de soutien pour naviguer efficacement dans un champ de recherche.

Pour répondre à cette problématique, la *génération de listes de lecture* vise à recommander des articles capables de guider les chercheurs dans la découverte d'un nouveau domaine (Ekstrand *et al.*,

2010 ; Gordon *et al.*, 2017). Ces approches s'intéressent à *quoi lire*, en produisant des listes d'articles liés à un sujet donné. Des travaux plus récents se sont intéressés à *comment lire*, en organisant les articles selon des structures reflétant l'évolution des thématiques ou les dépendances méthodologiques. Certaines méthodes ciblent spécifiquement l'ordonnancement des articles en optimisant des critères spécifiques (Sesagiri Raamkumar *et al.*, 2017 ; Figueira *et al.*, 2019) ; toutefois, ces critères sont définis de manière arbitraire et ne reflètent pas nécessairement les besoins réels des utilisateurs. D'autres travaux modélisent les relations de prérequis entre articles afin d'aider les nouveaux venus à identifier rapidement un ordre de lecture facilitant la compréhension de la littérature (Gordon *et al.*, 2017 ; Ding *et al.*, 2022). Dans les deux cas, l'ordonnancement repose sur des critères externes à l'utilisateur, tels que des attentes générales concernant une liste de lecture et son contenu, plutôt que sur une adaptation à ses besoins spécifiques. Une méthode proposée par Gordon *et al.* (2017) tente de modéliser les connaissances du lecteur à l'aide d'un formulaire utilisateur ; néanmoins, le système dépend toujours d'une entrée externe pour estimer le niveau du lecteur.

Une alternative prometteuse réside dans la recommandation séquentielle, une branche des systèmes de recommandation qui modélise l'historique d'interactions d'un utilisateur afin d'adapter ses prédictions à chaque individu (Boka *et al.*, 2024). Les approches notables dans ce domaine, tels BERT4rec (Sun *et al.*, 2019) ou SASRec (Kang & McAuley, 2018), se concentrent généralement sur la recommandation de produits et d'autres tâches orientées marketing. À notre connaissance, ces modèles n'ont pas été directement appliqués à la recommandation de documents scientifiques, possiblement à cause du manque de jeux de données de listes de lecture élaborées par des experts (Aubert-Béduchaud *et al.*, 2025).

Dans cet article, nous répondons à l'absence d'un cadre dédié à la recommandation séquentielle d'articles scientifiques en proposant une méthode automatique de construction de jeux de données à partir de collections existantes. Pour cela, nous formulons la recommandation séquentielle dans la littérature scientifique comme une tâche de prédiction du prochain article. Cette formulation permet non seulement de générer des parcours de lecture personnalisés, mais également d'entraîner les modèles directement à partir de la littérature elle-même, en utilisant les séquences de citations présentes dans les introductions comme traces naturelles de lecture.

Nous présentons un travail préliminaire consacré à la construction de la tâche de recommandation séquentielle appliquée à la littérature scientifique. Plus précisément, nous proposons un nouveau jeu de données dédié, ainsi qu'un protocole d'évaluation tenant compte des spécificités de cette tâche. Nous présentons également des résultats expérimentaux de référence, permettant de situer les performances de modèles de base sur ce problème.

# 2 Construction automatisée d'un jeu de données de séquences de lecture

## 2.1 Extraction de séquences de lecture

Afin de construire une séquence de lecture au sein d'un domaine de recherche, nous analysons les sections d'*introduction* des articles scientifiques à travers le prisme du modèle CARS (*Create A Research Space*) (Swales, 1990). Ce modèle décrit la progression rhétorique des introductions scientifiques, dans lesquelles le discours évolue progressivement d'un contexte général vers une

contribution scientifique spécifique. Dans cette optique, nous formulons les observations suivantes :

1. Puisque les auteurs structurent volontairement leur discours selon une structure en entonnoir, l'ordre des citations **peut capturer cette information de progression à travers la littérature citée**.
2. Lorsqu'une introduction d'article propose une synthèse d'un domaine de recherche, elle s'appuie généralement sur les travaux précédemment cités afin de restreindre progressivement la discussion vers le problème de recherche étudié, créant ainsi une **progression cohérente des idées dont l'article citant constitue l'aboutissement**.

Ces propriétés suggèrent que les séquences de citations comme extraites des introductions scientifiques reflètent l'organisation rhétorique du texte, en fournissant des points d'entrée cohérents dans un domaine de recherche et en facilitant naturellement la navigation dans la littérature. Nous étudions donc l'utilisation de ces séquences comme *proxy* pour des listes de lecture.

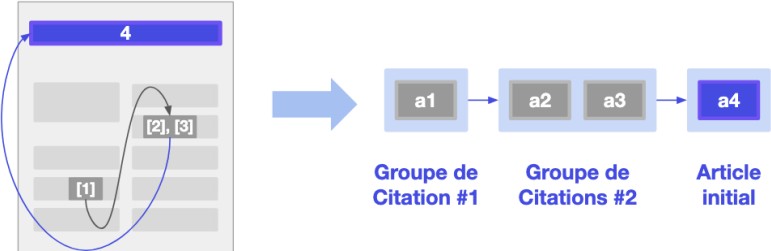

FIGURE 1 – Procédure d'extraction d'une séquence de lecture depuis un article initial.

Des travaux antérieurs ont montré que le contexte d'une citation fournit des indices importants sur sa fonction et son intention rhétorique (Teufel *et al.*, 2006; Abu-Jbara *et al.*, 2013), traduisant la propension des auteurs à regrouper des travaux connexes, tandis que les citations isolées jouent souvent des rôles argumentatifs plus spécifiques. Cette observation motive l'utilisation de groupes de multi-citations (une ou plusieurs citations appartenant à un même groupe de citations) comme cibles de prédiction, permettant au modèle de récupérer non seulement des références individuelles, mais aussi des ensembles cohérents de travaux liés. Nous proposons une méthode permettant de construire automatiquement une séquence de lecture favorisant le développement d'une expertise sur un sujet de recherche donné (voir figure 1). L'historique de lecture d'un domaine de recherche, extrait à partir d'un article initial, est représenté sous la forme d'une séquence ordonnée de groupes de multi-citations : pour chaque article d'un groupe, l'ensemble des articles du groupe suivant correspond à l'ensemble le plus probable de lectures ultérieures. Ainsi, pour tout article comportant une introduction, celle-ci est considérée comme un historique de lecture, et l'article lui-même comme la continuation de cet historique.

## 2.2 Collecte des données

Pour cette collecte de données, nous nous concentrons sur le domaine du Traitement Automatique des Langues (TAL) et utilisons l'ACL Anthology [1] comme source principale, couvrant un large

---

1. https://github.com/acl-org/acl-anthology

éventail de conférences et ateliers en TAL. Cette collection initiale est constituée des meta-données de 118 325 articles provenant de diverses conférences du domaine du TAL. À partir de ce corpus, nous sélectionnons ensuite les articles publiés dans les conférences et ateliers de l'Association for Computational Linguistics (ACL), dont le processus de relecture par les pairs permet une qualité de publication élevée.

Nous extrayons ensuite le contenu des PDFs des articles publiés aux événements ACL à l'aide de GROBID [2], outil permettant l'extraction structurée du contenu d'un article scientifique. Les sorties de GROBID peuvent être bruitées ou imprécises, nous alignons ainsi les citations extraites avec la collection ACL Anthology afin d'assurer leur validité et leur cohérence.

Nous considérons la première section des articles, correspondant essentiellement à l'introduction de l'article. Dans cette section, nous nous intéressons aux groupes de co-citations et leur contexte de citation dont nous extrayons le contenu et l'ordonnancement. Cette étape permet de constituer un premier artefact présentant le contenu bibliographique d'un ensemble d'articles ACL candidats.

Pour constituer les séquences de lecture, nous supprimons les doublons en ne considérant que la première occurrence d'une citation. Nous conservons les séquences de tailles comprises entre 3 et 20 articles scientifiques, dans la lignée des travaux liés aux listes de lecture scientifique (Aubert-Béduchaud *et al.*, 2025). Cette procédure permet l'identification de 30224 séquences de lectures potentielles.

## 2.3   Génération d'un besoin d'information

La procédure décrite ci-dessus permet l'extraction automatique des séquences de lecture de tout article présentant un historique de lecture dans son introduction. Néanmoins, cette stratégie ne permet pas directement d'identifier clairement le domaine de recherche de la séquence, limitant l'utilisation de celle-ci pour des tâches de prédiction.

En recherche d'information, le besoin d'information lié à un domaine se représente généralement sous la forme d'une requête d'information, souvent un ensemble de mots-clés ou une question en langage naturel (Pang & Kumar, 2011; White *et al.*, 2015).

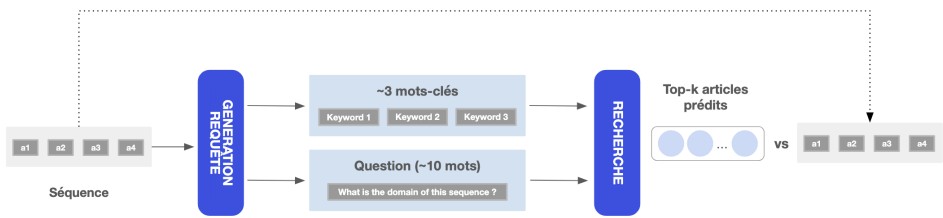

FIGURE 2 – Protocole d'annotation automatique du besoin d'information de séquences de lecture.

N'ayant pas accès aux requêtes initiales ayant conduit à la construction de la séquence, nous choisissons de les générer automatiquement à partir des séquences extraites. Afin de générer des requêtes initiales plausibles, nous cherchons à qualifier le domaine de recherche de la séquence par des requêtes courtes et génériques (voir figure 2). Nous nous appuyons sur des valeurs moyennes proposées dans

---

2. https://github.com/allenai/grobid (swh :1 :dir :324a18113b0c7624a66a21550bd0e8522e328b4e)

des collections de requêtes liées au domaine scientifique, à savoir environ 3 mots-clés (Boteva *et al.*, 2016 ; Singh *et al.*, 2023) et des questions en langage naturel d'environ 10 mots (Voorhees *et al.*, 2021 ; Nguyen *et al.*, 2022).

### 2.3.1 Modèles candidats

Nous explorons plusieurs paradigmes de modélisation pour la génération de requêtes à partir de séquences d'articles. Trois familles d'approches sont évaluées :

— Statistique (MultiPartiteRank) : extraction de mots-clés pour chaque article de la séquence, suivie d'une sélection par fréquence agrégée sur l'ensemble de la séquence.
— Expansion de document (docT5query) : génération de questions candidates pour chaque document, puis sélection de la plus représentative par similarité cosinus entre les plongements des questions et le plongement moyen de la séquence.
— Grands modèles de langue (GPT-4.1, Gemma3, Ministral3, Qwen3) : génération de mots-clés et de questions par conditionnement via instructions explicites.

L'ensemble des approches est paramétré de façon à produire environ 3 mots-clés ou une question d'environ 10 mots, selon le type de requête considéré.

### 2.3.2 Stratégie d'évaluation

Nous comparons les modèles sur la capacité des requêtes générées à retrouver les documents de la séquence lors du requêtage de la collection. Deux paradigmes de recherche sont employés : la recherche dense, via le modèle `sentence-transformer/allenai-specter`[3], et la recherche éparse, via BM25[4]. La sélection du meilleur modèle s'appuie conjointement sur les métriques R@k, MAP ainsi que sur la longueur des requêtes générées. Le jeu de données proposant un nombre important de documents, nous choisissons d'évaluer les tendances des modèles sur un sous-échantillon aléatoire de 1 000 séquences afin de limiter le coût en calculs.

Les résultats (présentés en table 1) mettent en évidence certaines tendances parmi les modèles évalués sur le sous-échantillon. Pour la génération de mots-clés, Ministral-3-8B obtient des résultats apparemment satisfaisants, mais produit un nombre moyen de mots-clés éloigné de notre cible. Cette quantité plus élevée de mots-clés ne correspondant pas à nos objectifs, nous écartons donc cette approche. MultiPartiteRank propose la même tendance dans une moindre mesure, avec un nombre de mots-clés légèrement excessif. Qwen3-4B apparaît comme le meilleur compromis entre les scores obtenus et la taille des requêtes. Pour la génération de questions, Qwen-4B présente des résultats intéressants, mais avec des requêtes souvent trop courtes. GPT-4.1-mini offre de bonnes performances générales, similaires à celles de Qwen-14B selon les métriques observées. Nous sélectionnons Qwen-14B pour des raisons de coûts lié à l'utilisation d'un modèle par API.

Les modèles sélectionnés sont ainsi **Qwen-4B pour les mots-clés** et **Qwen-14B pour la génération de questions**.

---

3. https://huggingface.co/sentence-transformers/allenai-specter
4. https://pyterrier.readthedocs.io

| | R@10 | | R@100 | | R@1000 | | MAP | | Taille | |
|---|---|---|---|---|---|---|---|---|---|---|
| | dense | éparse | dense | éparse | dense | éparse | dense | éparse | #mots | $|Q|$ |
| **Mots-clés** | | | | | | | | | | |
| multipartite rank | **0,052** | **0,188** | **0,116** | 0,337 | **0,263** | 0,520 | **0,033** | **0,135** | 3,981 | 32,828 |
| gpt-4,1-mini | 0,021 | 0,123 | 0,082 | 0,332 | **0,266** | 0,603 | 0,013 | 0,081 | 3,635 | 36,730 |
| gpt-4,1 | 0,015 | 0,134 | 0,065 | **0,360** | 0,222 | **0,631** | 0,009 | 0,083 | 3,086 | 31,420 |
| google/gemma-3-4b-it | 0,014 | 0,119 | 0,062 | 0,318 | 0,198 | 0,585 | 0,008 | 0,075 | 3,449 | 33,314 |
| google/gemma-3-12b-it | 0,014 | 0,130 | 0,054 | 0,341 | 0,206 | 0,599 | 0,008 | 0,089 | 3,163 | 30,585 |
| mistralai/Ministral-3-8B-Instruct-2512 | **0,048** | **0,180** | **0,149** | **0,388** | **0,364** | 0,624 | **0,029** | **0,119** | 5,324 | 49,911 |
| mistralai/Ministral-3-14B-Instruct-2512 | 0,019 | 0,130 | 0,083 | 0,325 | 0,245 | 0,577 | 0,012 | 0,079 | 3,834 | 38,572 |
| Qwen/Qwen3-4B | **0,025** | **0,167** | **0,086** | **0,372** | 0,245 | **0,619** | **0,015** | **0,112** | 3,616 | 33,365 |
| Qwen/Qwen3-14B | 0,015 | 0,143 | 0,068 | 0,339 | 0,217 | 0,594 | 0,009 | 0,091 | 3,188 | 32,093 |
| **Question** | | | | | | | | | | |
| castorini/doc2query-t5-base-msmarco | 0,059 | 0,174 | 0,135 | 0,306 | 0,297 | 0,492 | 0,040 | 0,121 | 6,050 | 37,470 |
| gpt-4,1-mini | 0,096 | **0,217** | **0,284** | **0,438** | **0,576** | **0,669** | 0,059 | 0,149 | 10,651 | 80,166 |
| gpt-4,1 | 0,080 | 0,190 | 0,247 | **0,422** | 0,538 | **0,653** | 0,048 | 0,128 | 10,108 | 74,797 |
| google/gemma-3-4b-it | 0,064 | 0,154 | 0,203 | 0,331 | 0,464 | 0,537 | 0,039 | 0,100 | 9,958 | 78,195 |
| google/gemma-3-12b-it | 0,074 | 0,181 | 0,232 | 0,380 | 0,506 | 0,615 | 0,043 | 0,124 | 10,205 | 76,688 |
| mistralai/Ministral-3-8B-Instruct-2512 | **0,098** | 0,208 | **0,280** | 0,409 | **0,555** | 0,624 | **0,060** | 0,148 | 10,991 | 82,692 |
| mistralai/Ministral-3-14B-Instruct-2512 | 0,087 | 0,211 | 0,262 | 0,415 | 0,540 | 0,629 | 0,054 | **0,152** | 10,893 | 82,999 |
| Qwen/Qwen3-4B | **0,111** | **0,252** | 0,268 | **0,438** | 0,520 | **0,630** | **0,068** | **0,190** | 8,123 | 65,210 |
| Qwen/Qwen3-14B | **0,101** | **0,232** | **0,274** | **0,422** | 0,532 | 0,621 | **0,061** | **0,170** | 10,439 | 74,941 |

TABLE 1 – Résultats consolidés par modèle et stratégie d'évaluation. R@k mesure le rappel parmi les k premiers résultats et MAP mesure la précision moyenne (meilleurs résultats de chaque métrique en gras). Les interprétations de longueur de requête sont déterminées par z-score mesurant leur écart à la longueur cible (écart faible , modéré , fort ).

### 2.3.3 Vérification de la cohérence

Nous cherchons à vérifier la cohérence des résultats à une échelle plus large et vérifier que les requêtes générées, tant mots-clés que questions, soient pertinentes en tant que requête initiale. Une particularité de la tâche est l'utilisation de séquences d'articles ordonnées dans la génération des requêtes. L'ordre des articles est un critère pouvant impacter les capacités des LLM sélectionnés à correctement représenter le domaine de recherche de la séquence Nous vérifions tout d'abord l'impact de la taille de la séquence initiale sur la génération de la requête d'information. Le premier élément de la séquence est en général très bien capturé par la requête. Toutefois, plus la séquence s'allonge, moins la requête générée parvient à identifier correctement les documents dans la collection. Un regain de performance est néanmoins observé pour le dernier document de la séquence. Cette tendance se manifeste à la fois pour les mots-clés et les questions générées et selon l'ordre de la séquence d'entrée du modèle (voir figure 3).

Nous mesurons pour cela la capacité des requêtes à retrouver les éléments de la séquence initiale. Pour cela, nous analysons les cas où les éléments d'une séquence ont été retrouvés dans le top 1 000 par l'une des deux approches. Nous retirons des données les échantillons où aucun élément de la séquence n'a été retrouvé par les termes de requête et nous marquons les requêtes capables d'identifier l'intégralité des éléments de la séquence (Table 2).

Cette procédure automatique permet ainsi l'extraction de **30 209** séquences de lecture issues d'articles publiés dans des conférences majeures ACL. Ces séquences sont accompagnées de requêtes courtes et longues caractérisant le besoin d'information initial associé à ces listes.

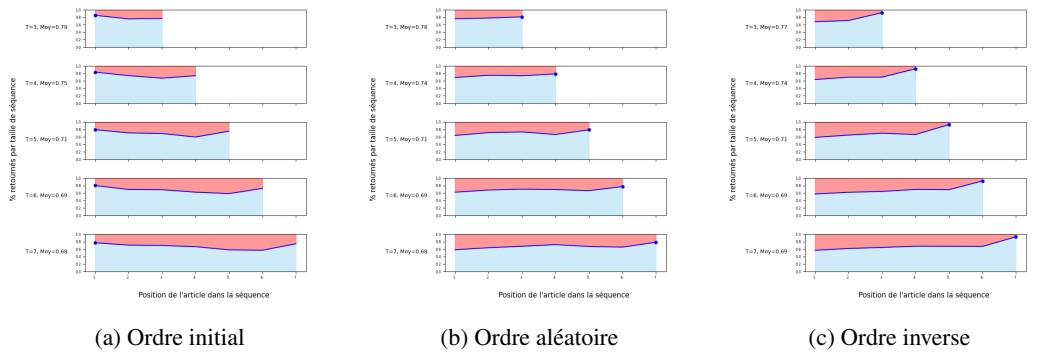

|            | (a) Ordre initial | (b) Ordre aléatoire | (c) Ordre inverse |
|------------|-------------------|---------------------|-------------------|

FIGURE 3 – Impact de l'ordre de la séquence d'entrée pour la génération de requêtes

|                                     | **Parfait** | **Partiel** | **Invalide** |
|-------------------------------------|-------------|-------------|--------------|
| Séquences                           | 13722       | 16487       | 15           |
| Nombre d'articles par séquence      | 4.45        | 6.75        | 3.93         |
| Nombre de mot-clés                  | 3.57        | 3.61        | 3.80         |
| Longueur des mot-clés (caractères)  | 32.82       | 33.62       | 34.93        |
| Longueur des question (mots)        | 10.57       | 10.34       | 10.80        |
| Longueur des question (caractères)  | 75.75       | 74.09       | 72.07        |

TABLE 2 – Statistiques liées à la cohérences des requêtes

# 3 Recommandation séquentielle d'articles scientifiques

## 3.1 Définition de la tâche

Nous proposons une nouvelle tâche de recommandation séquentielle adaptée à la recommandation d'articles scientifiques.

Soit $D$ un domaine de recherche, $Q_d$ une requête initiale liée à ce domaine formulée par un utilisateur, et $H_d = (a_1, \ldots, a_n)$ avec $a_i \in D$ son historique de lecture. La recommandation séquentielle vise à prédire le prochain article pertinent en combinant ces deux signaux :

$$a_{n+1}^* = \arg \max_{a \in D} \ P(a \mid Q_d, H_d).$$

## 3.2 Découpage temporel des données

Afin de garantir un minimum de chevauchement des données entre les ensembles d'entraînement, de validation et de test, nous adoptons un découpage temporel de nos données (Gusak *et al.*, 2025). L'idée est de n'inclure dans l'ensemble d'entraînement que les séquences antérieures à une certaine date, tandis que les ensembles de validation et de test contiennent les séquences postérieures à cette date. Le découpage temporel est ici appliqué à l'article cible associé à chaque séquence, de sorte

qu'aucune séquence de test ne puisse être contenue dans une sous-séquence d'entraînement (voir figure 4).

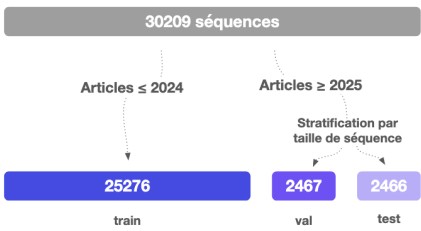

FIGURE 4 – Procédure de découpage du jeu de données en sous-ensembles expérimentaux

## 3.3 Protocole d'évaluation séquentielle

Le paradigme d'évaluation doit également tenir compte de la séquentialité de l'historique de lecture. Évaluer uniquement l'article final d'un historique de lecture reviendrait à ignorer les étapes intermédiaires qui ont permis la construction de la séquence. Pour cela, nous nous inspirons de la stratégie d'évaluation successive proposée par Gusak *et al.* (2025), que nous adaptons aux spécificités de nos données.

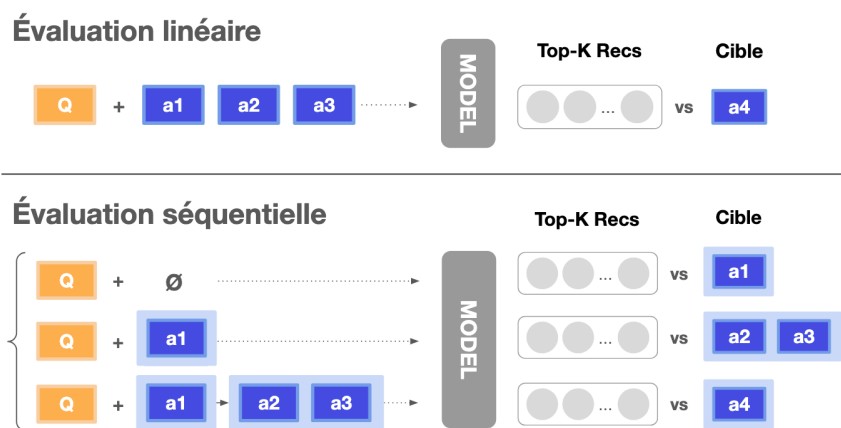

FIGURE 5 – Comparaison entre évaluation linéaire et évaluation séquentielle successive.

La figure 5 détaille le processus d'évaluation. Nous modélisons l'évaluation des couples de requêtes et d'historiques comme l'évaluation de chaque sous-séquence composant l'historique. Ce paradigme permet de mesurer les performances sur la tâche à chaque étape intermédiaire et permet de mieux refléter les capacités des modèles à prédire des articles de façon séquentielle.

# 4 Expérimentations

## 4.1 Modèles évalués

Nous évaluons les capacités des modèles BM25 et SPECTER2[5] à correctement réaliser la tâche de recommandation séquentielle d'article scientifique. La prédiction consiste en une interpolation du besoin d'information et de la moyenne du contenu de l'historique de lecture. L'objectif de cette interpolation est d'estimer de manière optimale le paramètre $\alpha$ sur un ensemble de validation, afin d'évaluer sur le jeu de test les performances de différentes approches exploitant soit uniquement le besoin d'information, soit l'historique de lecture, soit une combinaison de ces deux sources d'information.

Pour BM25, le vecteur de scores final $s$ est obtenu par interpolation entre les scores BM25 de la requête courante $s_{Q_q}$ et la moyenne des scores des requêtes de l'historique $s_{H_d} = \frac{1}{n}\sum_{i=1}^{n} s_{H_i}$, contrôlée par un paramètre $\alpha \in [0, 1]$ :

$$s = \alpha s_{H_d} + (1-\alpha)s_{Qd}, \qquad a_{n+1}^* = \arg\max_{a \in D} \ s(a).$$

Pour SPECTER2, le vecteur $\hat{q}$ permettant de requêter la collection est obtenu par interpolation entre l'embedding du besoin d'information $\hat{Q}_d$ et la moyenne des embeddings de l'historique $\hat{H}_d = \frac{1}{n}\sum_{i=1}^{n} \hat{a}_i$, contrôlée par un paramètre $\alpha \in [0, 1]$ :

$$\hat{q} = \alpha \hat{H}_d + (1-\alpha)\hat{Q}_d, \qquad a_{n+1}^* = \arg\max_{a \in D} \ \hat{q}^\top a$$

## 4.2 Résultats et discussion

Nous utilisons les valeurs de $\alpha$ maximisant les performances en termes de R@100 sur l'ensemble de validation pour l'évaluation sur le jeu de test, soit $\alpha = 0.2$ pour BM25 et $\alpha = 0.5$ pour SPECTER2 (voir annexe A). Nous considérons également deux configurations extrêmes afin d'analyser la contribution relative de chaque source d'information : une configuration reposant uniquement sur l'historique ($\alpha = 0$) et une autre exploitant exclusivement la requête ($\alpha = 1$).

| | R@1 | R@10 | R@100 | NDCG@100 |
|---|---|---|---|---|
| **Éparse** | | | | |
| BM25 Requête seule | 0,001 | 0,021 | 0,401 | 0,083 |
| BM25 Historique seul | 0,001 | 0,022 | 0,242 | 0,053 |
| BM25 Requête+Historique | **0,002** | **0,036** | **0,445** | **0,097** |
| **Dense** | | | | |
| SPECTER2 Requête seule | 0,000 | 0,014 | 0,307 | 0,063 |
| SPECTER2 Historique seul | 0,000 | 0,018 | 0,207 | 0,045 |
| SPECTER2 Requête+Historique | 0,001 | 0,022 | 0,375 | 0,079 |

(a) Mots-clés

| | R@1 | R@10 | R@100 | NDCG@100 |
|---|---|---|---|---|
| **Éparse** | | | | |
| BM25 Requête seule | 0,000 | 0,023 | 0,404 | 0,085 |
| BM25 Historique seul | 0,001 | 0,022 | 0,242 | 0,053 |
| BM25 Requête+Historique | 0,001 | **0,034** | **0,450** | **0,097** |
| **Dense** | | | | |
| SPECTER2 Requête seule | 0,000 | 0,017 | 0,365 | 0,075 |
| SPECTER2 Historique seul | 0,000 | 0,018 | 0,207 | 0,045 |
| SPECTER2 Requête+Historique | 0,001 | 0,025 | 0,410 | 0,086 |

(b) Questions

TABLE 3 – Résultats des modèles de référence sur le jeu de test.

Les résultats sont présentés dans la table 3. Le modèle BM25 obtient les meilleurs scores globaux sur cette tâche. Le fait que SPECTER2 affiche des performances inférieures à celles de BM25 pourrait

---

5. https://github.com/allenai/SPECTER2

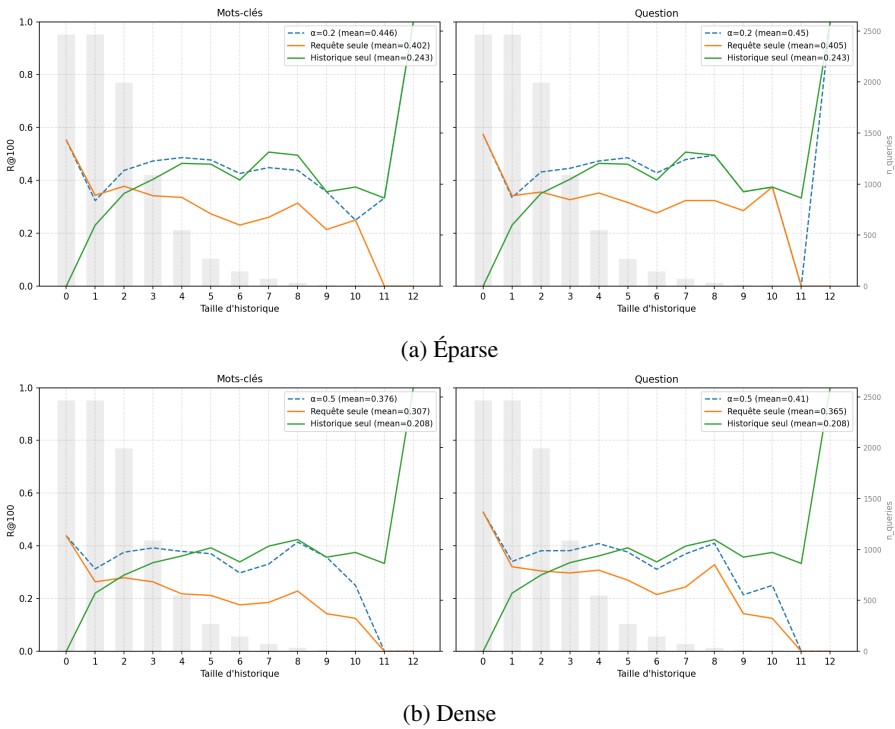

(a) Éparse

(b) Dense

FIGURE 6 – Étude de l'impact de l'historique selon sa taille

suggérer que les représentations sémantiques denses sont plus difficiles à exploiter dans ce contexte. Cela peut s'expliquer par l'utilisation de vecteurs issus du modèle SPECTER2 : celui-ci étant entraîné sur des triplets de citations scientifiques, il est possible que le contenu réel des articles ne soit pas toujours sémantiquement proche selon cette représentation.

Une analyse détaillée en fonction de la taille de l'historique met en évidence des tendances intéressantes (voir figure 6). Les requêtes initiales sont très performantes pour de faibles tailles d'historique, mais leurs performances diminuent fortement à mesure que l'historique s'enrichit. La perte entre un historique vide et un historique contenant un groupe de co-citations est nette. Pour des tailles d'historique importantes, la requête ne représente plus correctement le groupe de co-citation à prédire. À l'inverse, l'historique devient progressivement plus pertinent et dépasse systématiquement les performances des requêtes à partir d'une taille d'historique $\geq 2$.

Ces observations suggèrent que, si la requête est efficace pour identifier les premiers articles à lire, la suite du parcours de lecture s'inscrit davantage dans la continuité des références déjà consultées par l'utilisateur. Ceci motive l'entraînement de modèles dédiés à correctement représenter l'historique d'une séquence pour cette tâche. Également, ces résultats motivent l'exploration d'approches hybrides, couplant représentations lexicales et sémantiques, afin d'améliorer conjointement la couverture des candidats pertinents et l'intégration de l'historique de lecture.

# 5 Conclusion

Ces travaux présentent une approche préliminaire pour aborder la tâche de recommandation séquentielle appliquée à la littérature scientifique. Nous proposons un jeu de données dédié à la recommandation séquentielle d'articles scientifiques, construit à partir des séquences de citations présentes dans les sections d'introduction des articles. Cette approche offre une grande versatilité dans la construction de séquences de lecture ; pour ce travail, nous nous appuyons sur le corpus ACL pour générer ces séquences. Également, nous proposons un protocole d'évaluation tenant compte des spécificités de la tâche et des contraintes liées à la modélisation de la séquentialité. Finalement, nous évaluons plusieurs approches de base pour la résolution de la tâche et proposons différentes mesures de référence afin d'établir les performances des modèles initiaux.

Ces premiers résultats motivent l'entraînement de modèles dédiés ainsi que l'exploration d'approches hybrides, combinant représentations lexicales et sémantiques, afin d'optimiser à la fois la couverture des candidats pertinents et l'intégration de l'historique de lecture. Nous espérons que ces travaux initiaux encourageront la communauté à s'intéresser à ces approches, qui pourraient notamment permettre de proposer des références scientifiques plus pertinentes aux jeunes chercheurs, et ainsi accompagner leur progression dans un domaine de recherche.

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

# A   Étude de l'alpha

Nous estimons les valeurs optimales du paramètre d'interpolation $\alpha$ pour les deux paradigmes de modèles, en explorant l'intervalle $[0, 1]$ avec un pas de 0.1 sur le jeu de validation (voir figure 7).

Pour BM25, les performances dépendent fortement de la requête, suggérant une contribution limitée de l'historique. Ce comportement peut être attribué à la nature lexicale du modèle, qui rend l'exploitation conjointe de multiples interactions passées moins efficace dans un espace de grande dimension. À l'inverse, SPECTER2 tire davantage parti de l'historique, dont l'influence est comparable, voire supérieure, à celle de la requête sur plusieurs métriques. Ceci reflète une meilleure capacité à modéliser des contextes informationnels étendus. Les scores de SPECTER2 restent néanmoins en deçà de BM25 pour la majorité des configurations.

La baisse drastique des performances lors de l'utilisation exclusive de l'historique s'explique par la nature de notre protocole d'évaluation. La séquentialité étant modélisée, les étapes où l'historique est vide seront systématiquement mal représentées en l'absence de requête initiale.

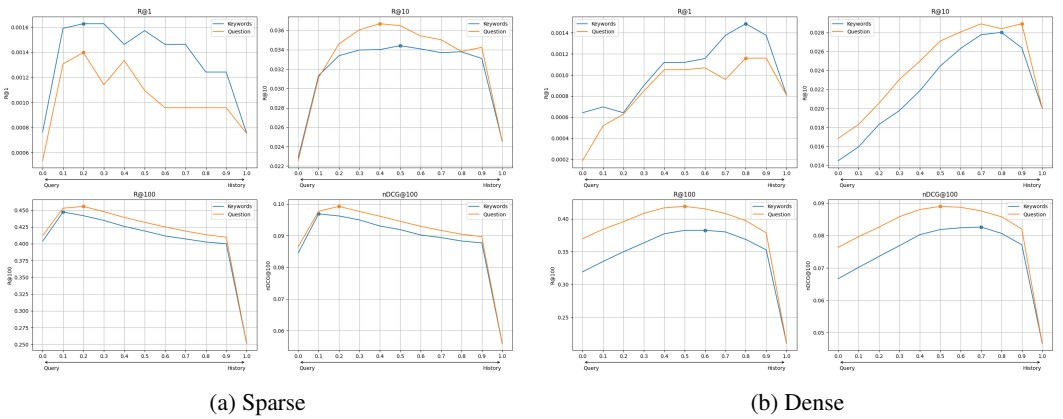

(a) Sparse                               (b) Dense

FIGURE 7 – Analyse de l'$\alpha$ d'interpolation optimal sur le jeu de validation.