# OpenReview forum: "Sequential Recommandation for Scholarly Articles"
_ls2n.fr/CORIA-TALN/2026/Workshop/ARTS — ls2n CORIATALN 2026 Workshop ARTS Submission_

### Official Review · Reviewer_i3PV · 2026-05-04

**Mode De Presentation:** Poster

**Confience:**

Oui

**Decision:**

Accepté

**Relecture:**

Forces

- La construction du jeu de données est bien décrite et les raisons justifiant les différentes étapes sont clairement exposées (par exemple, ils partent de l'hypothèse selon laquelle les introductions des articles listent des articles dans un certain ordre pédagogique qui peut être exploité).
- L'article explique clairement ce qui a été fait et présente une idée intéressante, même si les expériences reste pour l'instant préliminaires.

Faiblesses

- Certaines des idées qui sous-tendent la construction de l'ensemble de données sont, à mon avis, un peu trop simplifiées. Par exemple, je ne dirais pas que les co-citations sont nécessairement substituables : elles peuvent illustrer un même point tout en s'appuyant les unes sur les autres (c'est-à-dire qu'elles ont besoin des co-citations précédentes pour illustrer les points fondamentaux sur lesquels elles reposent). Cela mériterait une analyse distincte je pense afin de confirmer les hypothèses utilisées dans la construction du jeu de données. Plus important encore, j'ai quelques doutes quant à la qualité de certains des articles utilisés, notamment les plus récents (dans le jeu de test) qui pourraient souffrir de contenu « halluciné » par les LLM.
- L'article nécessite une relecture profonde, car il comporte de nombreuses fautes de grammaire et des tournures peu naturelles. De plus, les figures et les tableaux sont en anglais.


Commentaires/questions

- Tous les articles de l'anthologie de l'ACL sont-ils utilisés ? J'ai deux remarques à ce sujet : (i) la qualité des publications est très variable, et certains workshops en particulier ne garantissent pas nécessairement le niveau de qualité que l'article présent laisse entendre, (ii) les publications récentes peuvent être partiellement générées par l'IA (notamment pour les introductions et travaux connexes), ce qui pourrait clairement introduire un biais dans les données (ou des références fausses ou hallucinées). L'article « hallucitations » (https://arxiv.org/abs/2601.18724) montre qu'il s'agit d'un véritable problème, y compris dans les workshops affiliés à *ACL/EMNLP.
- Serait-il intéressant d'ajouter quelques références de la section « travaux connexes » ? Cette section reprend souvent le même fil conducteur que l'introduction, mais de manière plus détaillée ; il y a donc peut-être là un aspect à approfondir (en ajoutant des références plus spécifiques au sein des chaînes extraites de l'introduction).
- Je ne comprends pas très bien le commentaire concernant les résultats de Mistral pour la génération de mots-clés : pourquoi un nombre différent de mots-clés donnerait-il un bon score (est-ce bien ce que signifie cette phrase) ?
- Je ne suis pas très convaincue par la nature temporelle des divisions de l'ensemble de données (pour les raisons de qualité évoquées ci-dessus). Cela pourrait signifier que les articles du jeu de test (les plus récents) sont ceux de moindre qualité.

Typos, mise en forme, etc.

- Il y a beaucoup de fautes de grammaire (j'en répertorie quelques-unes ci-dessous, mais mes commentaires ne sont pas exhaustifs) et certaines tournures peu naturelles que je suggère de réviser
- Figure 6 -> figure 6 (en minuscules en français, de même pour les sections, les tableaux, etc.)
- Il y a une erreur dans le titre anglais (scholary → scholarly)
- Lecture. étant → point qui n'a pas sa place
- une introductions → une introduction
- Y a-t-il une référence pour GROBID ?
- en langage naturel. (White et al.) → point mal placé (ce n'est pas le seul exemple de cette erreur)
- automatiquedu → espace manquant
- (Singh et al., 2023 ; Boteva et al., 2016) -> Je m'attendrais à ce que ces références soient classées par ordre chronologique
- Ministral -> Mistral
- Requêtes générées… initial -> … générées… initiale
- Séquence initial -> Séquence initiale
- des modèles LLMs -> des LLM
- Les question -> les questions
- La note de bas de page 3 devrait être formatée comme un \url ou un \href
- Les tableaux et les figures sont en anglais - les figures devraient utiliser des virgules plutôt que des points et les en-têtes sont en anglais

**Resume:**

Cet article présente une nouvelle tâche qui consiste à prédire, de manière séquentielle, quel article de la littérature scientifique le lecteur devrait lire, en fonction de son expérience et de ce qui a été prédit comme lecture précédente. Cela contraste avec la plupart des études sur la recommandation d'articles, qui consistent à prédire des listes statiques non personnalisées. Les auteurs présentent un travail préliminaire qui décrit la construction des données, un protocole d'évaluation de la tâche et des expériences de benchmarking (BM25 et SPECTER2).

---

### Official Review · Reviewer_s3Hg · 2026-05-06

**Mode De Presentation:** Poster

**Confience:**

Oui

**Decision:**

Accepté

**Relecture:**

La tâche est sans aucun doute pertinente et intéressante.

L’approche est cependant déroutante. En effet, une séquence de référence est automatiquement extraites à partir d’article initiaux, en formulant l’hypothèse que "pour tout article scientifique, il est possible de construire une séquence d’articles constituant un chemin optimal de lecture pour monter en compétence sur le sujet de recherche traité à partir de son introduction". La validité de cette hypothèse, notamment quant à la séquentialité, devrait a minima être évaluée à l’aide de tests utilisateurs.

De plus, l’outil utilisé pour extraire ces séquences (Grobid) n’est pas infaillible. Le choix de faire de ces séquences automatiquement extraites une référence semble largement discutable. Il semble que rien, dans les expériences proposées, ne valide l’ordre de ces séquences.

**Resume:**

L’article introduit une nouvelle tâche : la recommandation, à partir d’un historique de lecture, d’une liste d’articles à lire séquentiellement.
La construction d’un jeu de données dédié à la tâche est décrite. Un protocole d’évaluation est détaillé. Un certain nombre de modèles de base pour résoudre le problème sont testés.

---

### Decision · Program_Chairs · 2026-05-07

Accept (Poster)